# Robust Bain distortion in the premartensite phase of a platinum-substituted Ni$_2$MnGa magnetic shape memory alloy

Sanjay Singh[1,2], B. Dutta [3], S.W. D'Souza[1], M.G. Zavareh[1], P. Devi[1], A.S. Gibbs[4], T. Hickel[3], S. Chadov[1], C. Felser[1] & D. Pandey[2]

The premartensite phase of shape memory and magnetic shape memory alloys (MSMAs) is believed to be a precursor state of the martensite phase with preserved austenite phase symmetry. The thermodynamic stability of the premartensite phase and its relation to the martensitic phase is still an unresolved issue, even though it is critical to the understanding of the functional properties of MSMAs. We present here unambiguous evidence for macroscopic symmetry breaking leading to robust Bain distortion in the premartensite phase of 10% Pt-substituted Ni$_2$MnGa. We show that the robust Bain-distorted premartensite (T2) phase results from another premartensite (T1) phase with preserved cubic-like symmetry through an isostructural phase transition. The T2 phase finally transforms to the martensite phase with additional Bain distortion on further cooling. Our results demonstrate that the premartensite phase should not be considered as a precursor state with the preserved symmetry of the cubic austenite phase.

[1] Max Planck Institute for Chemical Physics of Solids, Nöthnitzer Str. 40, D-01187 Dresden, Germany. [2] School of Materials Science and Technology, Indian Institute of Technology (Banaras Hindu University), Varanasi 221005, India. [3] Max-Planck-Institut für Eisenforschung Max-Planck-Strasse 1, 40237 Düsseldorf, Germany. [4] ISIS Pulsed Neutron Facility, STFC, Rutherford Appleton Laboratory, Chilton, Didcot, Oxfordshire OX11-0QX, UK. Correspondence and requests for materials should be addressed to S.S. (email: sanjay.singh@cpfs.mpg.de)

The appearance of a precursor state, widely known as the premartensite (PM) phase, is an interesting feature of compounds/alloys exhibiting martensitic phase transitions. This has been extensively studied in conventional shape memory alloys (SMAs)[1–9]. The PM state appears in between the parent austenite and martensite phases with preserved parent phase symmetry (cubic) in the sense that the austenite peaks in the diffraction patterns do not exhibit any signature of symmetry breaking or in other words any evidence for Bain distortion. The characteristic signatures of the premartensite state are the appearance of diffuse streaks and superlattice spots in the diffraction patterns[10, 11].

Another important class of alloys that owe their functional properties to a martensitic phase transition are magnetic SMAs (MSMAs), where shape recovery can be realized under the influence of not only the temperature and stress but also the magnetic field. Among the MSMAs, stoichiometric $Ni_2MnGa$ and off-stoichiometric Heusler alloys in the alloy systems Ni–Mn–X (Ga, In, Sn, Sb) have received considerable attention[12–21]. The stoichiometric $Ni_2MnGa$ system is by far the most studied system due to its large (10%) magnetic field-induced strain (MFIS) with potential for applications in novel multifunctional sensor and actuator devices[13, 14, 22]. The recent discovery of skyrmions in $Ni_2MnGa$ has opened newer potential applications for this material in the field of spintronics as well[23]. $Ni_2MnGa$ is a multiferroic SMA with two ferroic order parameters, namely spontaneous magnetization and spontaneous strain, which appear below the ferromagnetic and ferroelastic (martensite) phase transition temperatures $T_C \sim 370$ K and $T_M \sim 210$ K[24, 25], respectively. The martensitic transition in $Ni_2MnGa$ is preceded by a PM phase transition around $T \sim 260$ K[26, 27]. Both the martensite and PM phases exhibit very strong coupling between the two ferroic order parameters (i.e., magnetoelastic coupling) which is responsible for the large MFIS. The large MFIS is also closely linked with the existence of a long period modulated structure of the martensite phase in the $Ni_2MnGa$ alloy[26, 28–30]. Since the modulated phase of $Ni_2MnGa$ appears via a modulated PM phase and not directly from the austenite phase after its Bain distortion, understanding the characteristics of the PM phase and its effect on the martensite phase transformation has been a topic of intense research in recent years[12, 20, 25, 29–38]. The austenite-PM and the PM-martensite transitions are first order in nature, with characteristic thermal hysteresis and phase coexistence regions[27], due to the strong magnetoelastic coupling[12] between the magnetization and strain caused by a soft $TA_2$ phonon mode at a wave vector $q \sim 1/3$[31, 33, 39–41]. The nature of the modulation in the PM and martensite phases has been shown to be 3 M and 7M-like but incommensurate[25, 27, 42, 43]. The incommensurately modulated PM and the martensite phases coexist over a wide temperature regime across $T_M$[27] suggesting that the martensite phase in $Ni_2MnGa$ originates from the PM phase through a first-order phase transition. While the martensite phase of $Ni_2MnGa$ exhibits macroscopic symmetry breaking[26, 44], the PM phase does not exhibit any evidence of symmetry breaking transition.

In the phenomenological theories of martensite phase transition, the transformation occurs through a lattice deformation shear, with atomic shuffles if necessary, leading to Bain distortion of the austenite unit cell while maintaining an invariant habit plane (contact plane) between the parent and the martensite phases achieved through a lattice invariant shear produced by twinning or stacking faulting[45]. Obviously, it would have been kinetically more favorable if the austenite phase had first undergone a mild Bain distortion in the PM phase that could have provided a further pathway to the martensite phase transformation with higher Bain distortion while maintaining the integrity of the habit plane all along. In the absence of any

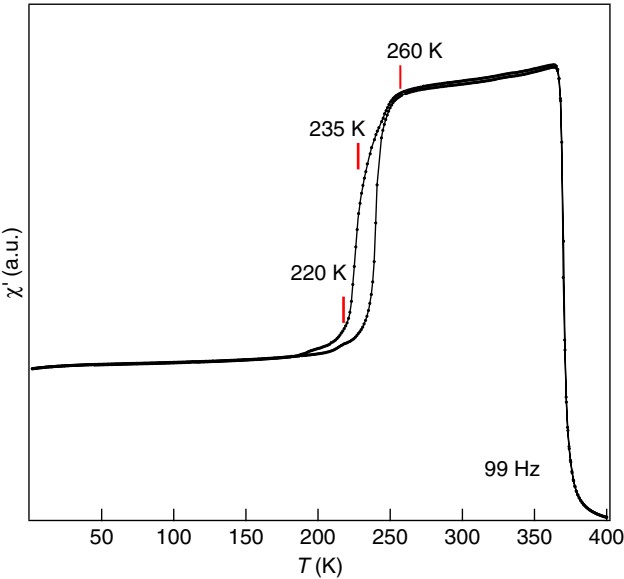

**Fig. 1** Temperature-dependent ac-susceptibility. Real part of ac-$\chi$ as a function of temperature in cooling and heating cycles. Red ticks indicate the temperatures that correspond to the different phases observed in synchrotron X-ray powder diffraction (see text)

macroscopic Bain distortion, the PM phase of $Ni_2MnGa$ does not seem to have a role in facilitating the appearance of Bain distortion in the martensite phase with an invariant habit plane. This then raises a fundamental question as to why at all the PM phase arises in $Ni_2MnGa$. Interestingly, first principles calculations reveal a local minimum that corresponds to an intermediate value of the Bain distortion in the PM phase[46]. However, this has not been verified experimentally till now.

We present here experimental evidence for macroscopic symmetry breaking in the PM phase of $Ni_2MnGa$ as a result of 10% Pt substitution leading to a robust Bain distortion. From temperature-dependent high resolution synchrotron X-ray powder diffraction (SXRPD) data analysis, we show that the Bain distortion grows in steps. First, a nearly 3 M modulated PM phase (PM(T1)) with an average cubic-like feature (i.e., negligible Bain distortion) of the elementary $L2_1$ unit cell results from the austenite phase. This phase then undergoes an isostructural phase transition to another 3M-like PM phase (PM(T2)) with robust Bain distortion at lower temperatures. Our results reveal that the two PM phases are stable thermodynamic phases and not precursor phases with preserved austenite symmetry, as has been presumed in the literature all along. Further, we also show that the martensite phase originates from the larger Bain-distorted PM phase suggesting that the Bain distortion appears in steps to facilitate the emergence of an invariant habit plane. These observations provide us an opportunity to critically examine the two existing mechanisms of the origin of modulation in these systems.

## Results

**Temperature-dependent ac-susceptibility**. The temperature ($T$) dependence of the ac-susceptibility (ac-$\chi$) of the $Ni_{1.9}Pt_{0.1}MnGa$ powder sample shown in Fig. 1 reveals a sharp change at paramagnetic-ferromagnetic transition temperature $T_c \sim 370$ K. On further cooling, the ac-$\chi$ decreases with a change of slope in the $\chi$–$T$ plot around $T \sim 260$ K. Interestingly, there is another change of slope in ac-$\chi$ around $T = 235$ K preceded by a large (~40%) drop in ac-$\chi$. In general, a large drop in ac-$\chi$ (or dc magnetization in low field) in the Ni-Mn-Ga based MSMAs is

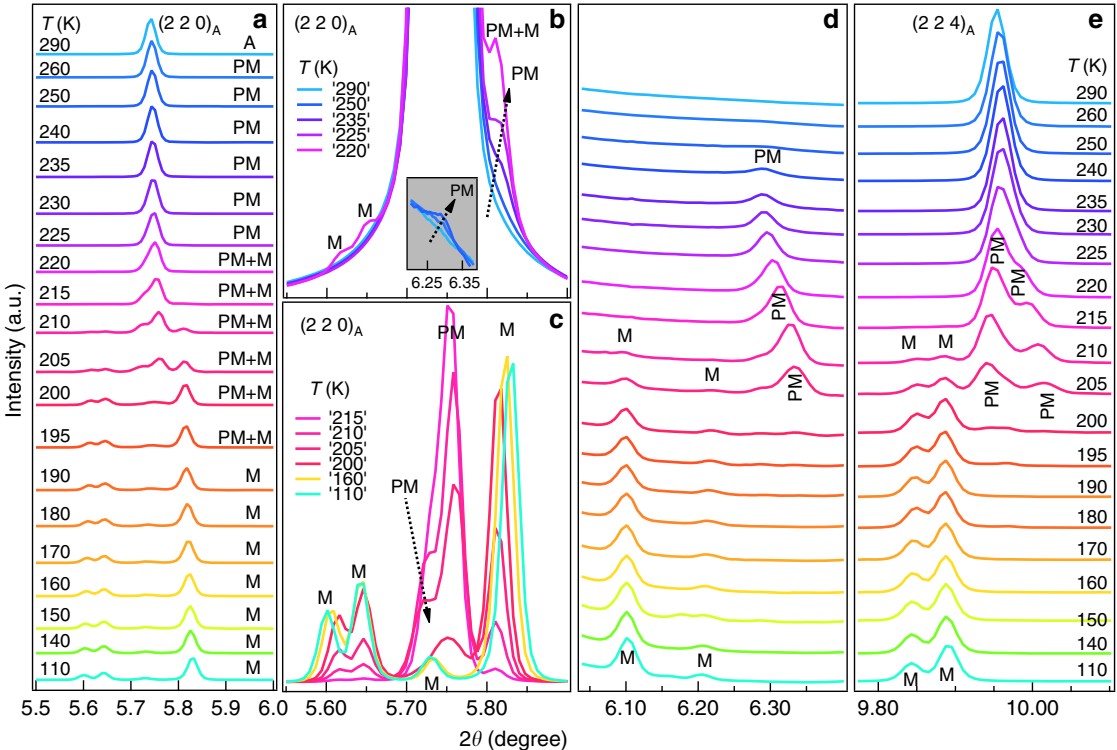

**Fig. 2** Temperature-dependent synchrotron X-ray powder diffraction patterns. Evolution of synchrotron X-ray powder diffraction patterns of $Ni_{1.9}Pt_{0.1}MnGa$ as a function of temperature during a cooling cycle for 3 selected $2\theta$ ranges. The (220) austenite peak region is shown in **a**, the satellite region in **d**, and the (224) austenite peak region in **e**. **b**, **c** represent the (220) austenite peak region with an expanded scale for selected temperatures. A, PM, and M represent the Bragg peaks of the cubic austenite, premartensite, and martensite phases, respectively. The inset in **b** shows the Bragg reflections at the extended scale in the $2\theta$ range between 6.20 and 6.38 where the satellite reflection of 3 M modulated premartensite phase is expected

related with the large magnetocrystalline anisotropy of the martensite phase[47, 48] but Fig. 1 suggests that nearly 40% of the total drop in ac-$\chi$ has already occurred at 235 K in the PM phase as a result of 10% Pt substitution. It is important to mention here that while in $Ni_2MnGa$ a very small (nearly 0.5%) drop in ac-$\chi$ at the PM phase during cooling from the austenite phase is observed[27] for $Ni_{1.9}Pt_{0.1}MnGa$ this drop is very large, which indicates a larger magnetocrystalline anisotropy of the PM phase in $Ni_{1.9}Pt_{0.1}MnGa$.

**High resolution synchrotron X-ray powder diffraction**. Our high resolution SXRPD patterns show clear correlation between the change of slope in the $\chi$–$T$ plot and structural changes corresponding to the PM and martensite phases discussed above. We show in Fig. 2 the variation of SXRPD patterns in selected $2\theta$ ranges with temperature in the 290–110 K range which captures the entire sequence of phase transitions. The SXRPD pattern at 290 K corresponds to the cubic austenite phase as it does not show any splitting of the XRD peaks. The Rietveld refinement using the 290 K data for the cubic structure ($Fm$-$3m$ space group) shows excellent fit between the observed and calculated peak profiles which further confirms the cubic structure at this temperature (Fig. 3a). The refined cell parameter $a = 5.84753(6)$ Å obtained from the Rietveld refinement is slightly higher than that of the stoichiometric $Ni_2MnGa$ ($a = 5.82445(1)$ Å), which might be due to the larger atomic size of the Pt (1.77 Å) atoms compared to the Ni (1.37 Å) atoms. This is in good agreement with an earlier study[49]. On cooling the sample below $T = 260$ K, several smaller intensity peaks appear. Two such peaks are marked in Fig. 2b and the inset of Fig. 2b. The low intensity peaks are the satellites that appear due to the modulated nature of the PM phase in $Ni_2MnGa$[27]. The intensity of these satellites increases

without much effect on the austenite cubic peaks on further cooling up to $T = 240$ K. The situation is similar to that in $Ni_2MnGa$, where cooling below the PM phase transition temperature ($T_{PM} = 261$ K) leads to the appearance of low intensity peaks. On the other hand, the austenite peaks remain nearly unaffected and retain their cubic nature[27]. The SXRPD of $Ni_{1.9}Pt_{0.1}MnGa$ in the temperature range 260–240 K thus looks similar to the modulated structure of the PM phase of $Ni_2MnGa$, which is incommensurate with 3M-like modulation[25–28, 30, 50, 51]. Therefore, we carried out Rietveld analysis of the 240 K SXRPD pattern of $Ni_{1.9}Pt_{0.1}MnGa$ using the (3 + 1) D superspace group approach taking into account both the main and the satellite reflections. We use the modulated structure of the PM phase similar to that of $Ni_2MnGa$ with the superspace group $Immm$ (00$\gamma$) s00. A fit between the observed and calculated peak profiles can be seen in Fig. 3b. The refined lattice parameters at 240 K are found to be: $a = 4.1337(2)$ Å, $b = 5.8416$ (3) Å, and $c = 4.1325$ (1) Å with a modulation wave vector (**q**) of $\mathbf{q} = 0.325\mathbf{c}^\star = (1/3-\delta)\,\mathbf{c}^\star$, where $\delta = 0.0083$ is the degree of incommensuration of modulation at 240 K. The magnitude of **q** confirms the incommensurate nature of the modulation. The closest rational approximant to the observed value is $(1/3)\mathbf{c}^\star$ which suggests 3M-like modulation in the PM phase. Thus the modulated structure of the PM phase of $Ni_{1.9}Pt_{0.1}MnGa$ stable in the temperature range 260–240 K, is identical to the PM phase of $Ni_2MnGa$. The refined lattice parameters and modulation vector are also close to the PM phase of $Ni_2MnGa$[25].

At ~235 K, which is close to the slope change in the ac-$\chi$ data (marked with the tick in Fig. 1), an interesting feature is observed in the diffraction pattern. At this temperature, the splitting of the Bragg peaks of the cubic austenite structure begins to show (Fig. 2d, e). However, no additional/new satellite reflections

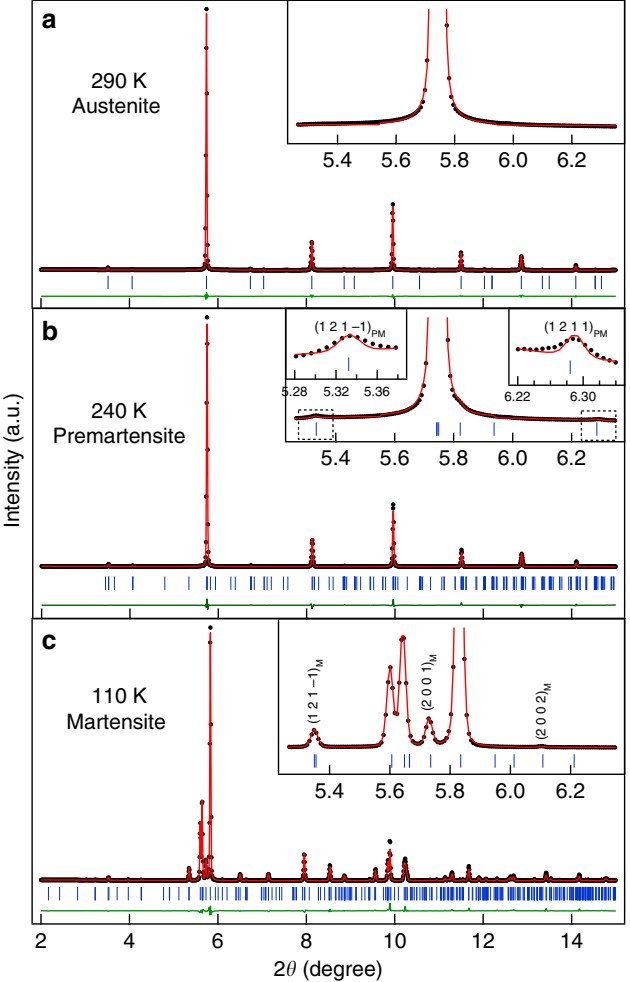

**Fig. 3** Rietveld fits for the synchrotron X-ray powder diffraction patterns of $Ni_{1.9}Pt_{0.1}MnGa$. Rietveld refinements at **a** 290 K (cubic austenite phase), **b** 240 K (premartensite phase), and **c** 110 K (martensite phase). The experimental data, fitted curve, and the residue are shown by circles (black), red continuous line, and green continuous line, respectively. The tick marks (blue) represent the Bragg peak positions. The insets show the fit for the main peak region on an expanded scale. Satellite reflections in the premartensite (PM) and martensite phases (M) are shown by $(hklm)_{PM}$ and $(hklm)_M$, respectively

appear. This structure therefore corresponds to the PM phase only. The satellite peaks are marked with PM in Fig. 2b. Later, we will show that this structure is a new PM phase having the same 3M-like incommensurate modulation but with a robust Bain distortion. This phase is observed up to 225 K while the splitting of cubic austenite peaks continues to increase from 235 K down to 225 K. At $T \sim 220$ K, new peaks have started appearing (marked with M in Fig. 2c–e). This is similar to the situation in $Ni_2MnGa$, where new satellite peaks of martensite phase appear after the PM phase is cooled below $T_M$[27]. The intensity of these new peaks grows at the expense of the PM phase peaks up to 190 K below which the PM phase peaks (both main and satellites) disappear completely. Below this temperature (190 K), no additional peaks appear in the structure down to the lowest temperature of our measurement 110 K. Thus it can be safely concluded that the martensite structure is stable down to the lowest temperature of measurements (110 K) and even below this temperature as confirmed by our latest neutron scattering measurements.

We investigated the structure of the martensite phase at 110 K by Rietveld refinement in the (3 + 1) D superspace similar to that in $Ni_2MnGa$. The main peaks originating from the splitting of the main austenite peaks could be indexed using an orthorhombic unit cell and space group *Immm* as in case of $Ni_2MnGa$. The unit cell parameters obtained after Le Bail refinement were found to be $a = 4.2390$ (1) Å, $b = 5.5682$ (1) Å, and $c = 4.207412$ (1) Å. The Rietveld refinement was then carried out using the superspace group *Immm* (00γ) s00 by taking into account the complete diffraction pattern including both the main and satellite reflections. The good fit between the observed and calculated profiles (Fig. 3c) confirms that the refinement has converged successfully. The refined value of the incommensurate modulation vector was found to be $\mathbf{q} = 0.4290$ (2) $\mathbf{c}^\star = (3/7 + \delta) \, \mathbf{c}^\star$, where $\delta = 0.00112$ is the degree of incommensuration. Thus the structure at 110 K (which is a representative of the martensite phase stable below 220 K) shows incommensurate modulation similar to the martensite phase of $Ni_2MnGa$[27, 28, 42, 52], which is of nearly 7M type.

The structures of the cubic austenite, PM in the temperature range 260–240 K and martensite at 110 K, which is representative of the structure below 220 K of $Ni_{1.9}Pt_{0.1}MnGa$, are similar to $Ni_2MnGa$. However, the structure of the PM phase of $Ni_{1.9}Pt_{0.1}MnGa$ in the temperature range 235–195 K, over which the splitting of the cubic austenite peaks is observed without the presence of the martensite peaks, is quite unusual and we now proceed to analyze this newly observed phase. The splitting of the austenite peaks in the PM phase indicates the loss of pseudocubic symmetry of the basic structure. This is unexpected for the PM phase as it is always considered to be a micromodulated phase with preserved cubic symmetry of the austenite peaks. We consider the 225 K data as a representative of this new structure and present the result of fits between the calculated and observed intensities by Rietveld refinement. As in the case of the Rietveld refinement of the PM and martensite phases stable in the temperature ranges 245 K < T < 260 K and T < 225 K, respectively we first considered only the main reflections (not satellites) and carried out Le Bail refinement using the cubic *Fm-3m* space group. From Fig. 4a, it can be clearly seen that this model misses out the new peaks, which result due to the splitting of the cubic austenite peaks (see the insets of Fig. 4a). This confirms that the cubic symmetry is now broken. We therefore considered the tetragonal unit cell with space group *I4/mmm*, which could capture the splitting but still the observed and calculated peak profiles are not matched properly. Finally, we took orthorhombic distortion into account in the space group *Immm*, which was also used for the martensite phase. The excellent fit between observed and calculated peak profiles (Fig. 4c and inset) shows that at 225 K the basic structure is orthorhombic with refined lattice parameters as: $a = 4.1356(2)$ Å, $b = 5.8276(2)$ Å, and $c = 4.1371$ (1) Å. There is a clear evidence for the pseudo-tetragonal Bain distortion of the PM phase with $b/\sqrt{2}a \sim 0.9965$. Having established the Bain-distorted orthorhombic nature of the PM phase at 225 K, we then carried out Rietveld refinement considering the complete diffraction pattern including satellite reflections using superspace group *Immm* (00γ) s00. An excellent fit between the observed and calculated peak profiles was obtained after the Rietveld refinements (Fig. 4d and inset). The refined modulation wave vector $\mathbf{q} = 0.3393 \, \mathbf{c}^\star$ shows incommensurate nature of the modulation with $\mathbf{q} = (1/3 + \delta) \, \mathbf{c}^\star$, where $\delta = 0.0059$ is the degree of incommensuration. Our results using high resolution SXRPD data thus provide evidence for a new incommensurately modulated 3M-like PM phase with broken cubic austenite symmetry. This is the example of macroscopic symmetry breaking and Bain distortion in the PM phase of any Heusler MSMA.

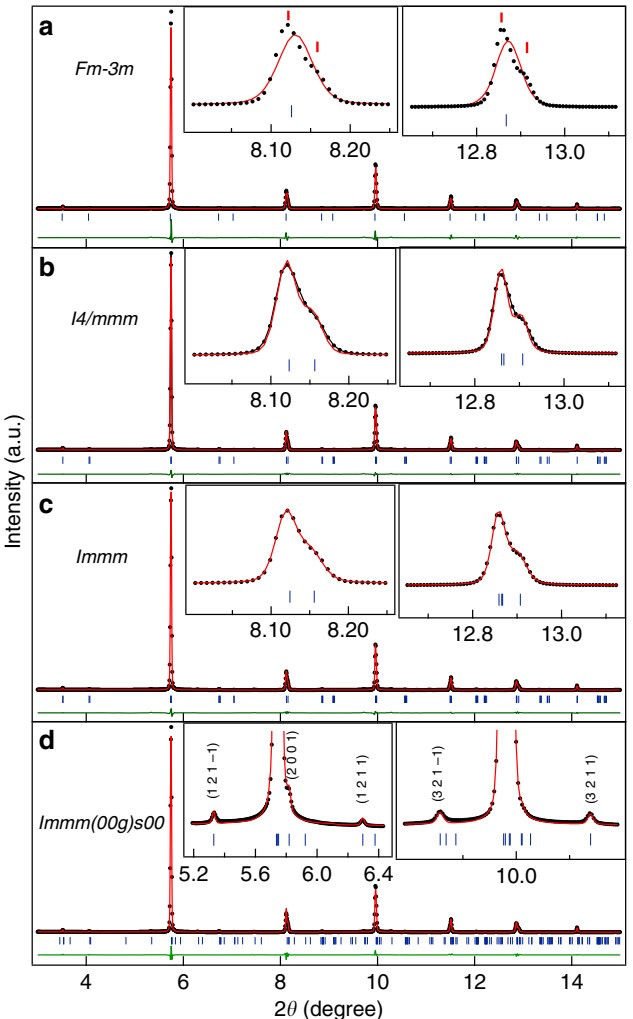

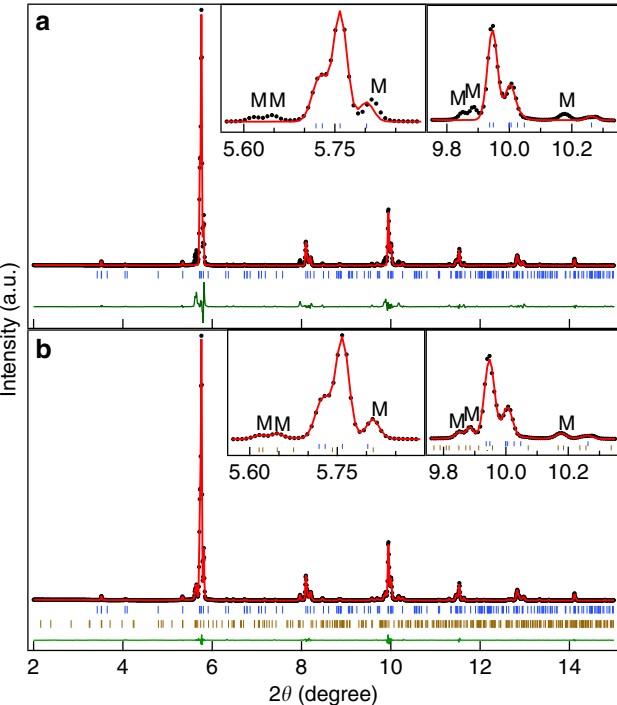

**Fig. 4** Rietveld fits for the synchrotron X-ray powder diffraction patterns for main peaks at 225 K. Rietveld refinements with **a** cubic austenite cell **b** Bain-distorted tetragonal unit cell **c** Orthorhombic unit cell. **d** Rietveld fits with 3 M modulated incommensurate premartensite structure. The experimental data, fitted curve, and the residue are shown by circles (black), red continuous line, and green continuous line, respectively. The tick marks (blue) represent the Bragg peak positions. The insets in **a**–**c** show the fit for the main reflections and in **d** for satellite reflections corresponding to the premartensite (PM)

**Fig. 5** Rietveld fits for the synchrotron X-ray powder diffraction patterns at 210 K. Rietveld refinements with **a** 3 M incommensurate premartensite structure and **b** Combination of 3 M premartensite and 7 M martensite structure. The experimental data, fitted curve, and the residue are shown by circles (black), red continuous line, and green continuous line, respectively. The tick marks (blue) represent the Bragg peak positions for the premartensite phase and the lower ticks (dark yellow) in **b** represents the Bragg peak positions for the martensite phase. The insets show the Bragg reflections on an expanded scale

After observing a different type of the PM phase with robust Bain distortion, we now proceed to present evidence for the coexistence of the incommensurate 3M-like Bain-distorted PM and incommensurate 7M-like martensite phases below 220 K. For this, we show in Fig. 5 the results of Rietveld refinement using the SXRPD pattern at 210 K. It can be observed from the inset of Fig. 5a that consideration of only the 3M-like incommensurate PM phase cannot account for some of the Bragg reflections (marked with M in the inset of Fig. 5a). We also verified that single phase 7M-like incommensurate martensite phase structure cannot account for this diffraction pattern. However, a refinement based on coexistence of both 3M-like Bain-distorted PM as well as 7M-like martensite phases gives an excellent fit between the observed and calculated profiles accounting for all of the peaks (Fig. 5b). From this, we conclude that the Bain-distorted 3M-like PM and 7M-like martensite phases, both with incommensurate modulations, coexist at 210 K. Using superspace Rietveld refinement, the coexistence of 3M-like and 7M-like

incommensurate structures was confirmed in the entire temperature range from 210 to 195 K. As phase coexistence is a typical characteristic of a first-order phase transition, our result show that the Bain-distorted PM phase to martensite phase transition is a first-order transition. Hence, the martensite phase originates from the PM phase in $Ni_{1.9}Pt_{0.1}MnGa$ similar to that in $Ni_2MnGa$[27]. However, there is a major difference in the case of $Ni_{1.9}Pt_{0.1}MnGa$ since the PM phase from which the martensite phase results shows robust Bain distortion in marked contrast to $Ni_2MnGa$ where the martensite phase results from a PM phase that preserves the cubic symmetry of the main unit cell.

Figure 6a shows the variation of the lattice parameters and unit cell volume for the equivalent cubic, PM ($a_{pm} \approx (1/\sqrt{2}) a_c$, $b_{pm} \approx a_c$, $c_{pm} \approx (1/\sqrt{2}) a_c$) and martensite ($a_m \approx (1/\sqrt{2}) a_c$, $b_m \approx a_c$, $c_m \approx (1/\sqrt{2}) a_c$) with temperature. It is evident from Fig. 6a that unit cell volume for all the phases decreases with decreasing temperature. However, the volume of the cubic austenite phase changes rather smoothly across the austenite to PM (T1) phase transition (260–245 K), similar to that in $Ni_2MnGa$, as expected for a weak first-order phase transition. However, on lowering the temperature further, a discontinuous change in the volume is observed at $T \sim 235$ K with a concomitant splitting of the cubic austenite peaks (Figs. 6a and 2e) and change of slope in the ac-$\chi$ vs. $T$ plot (Fig. 1). This discontinuous change in volume occurs at the cubic premartensite to the Bain-distorted PM phase transition and confirms the first-order nature of this phase transition. Furthermore, a clear discontinuous volume change during Bain-distorted PM (T2) to martensite phase transition is also observed confirming the first-order character of

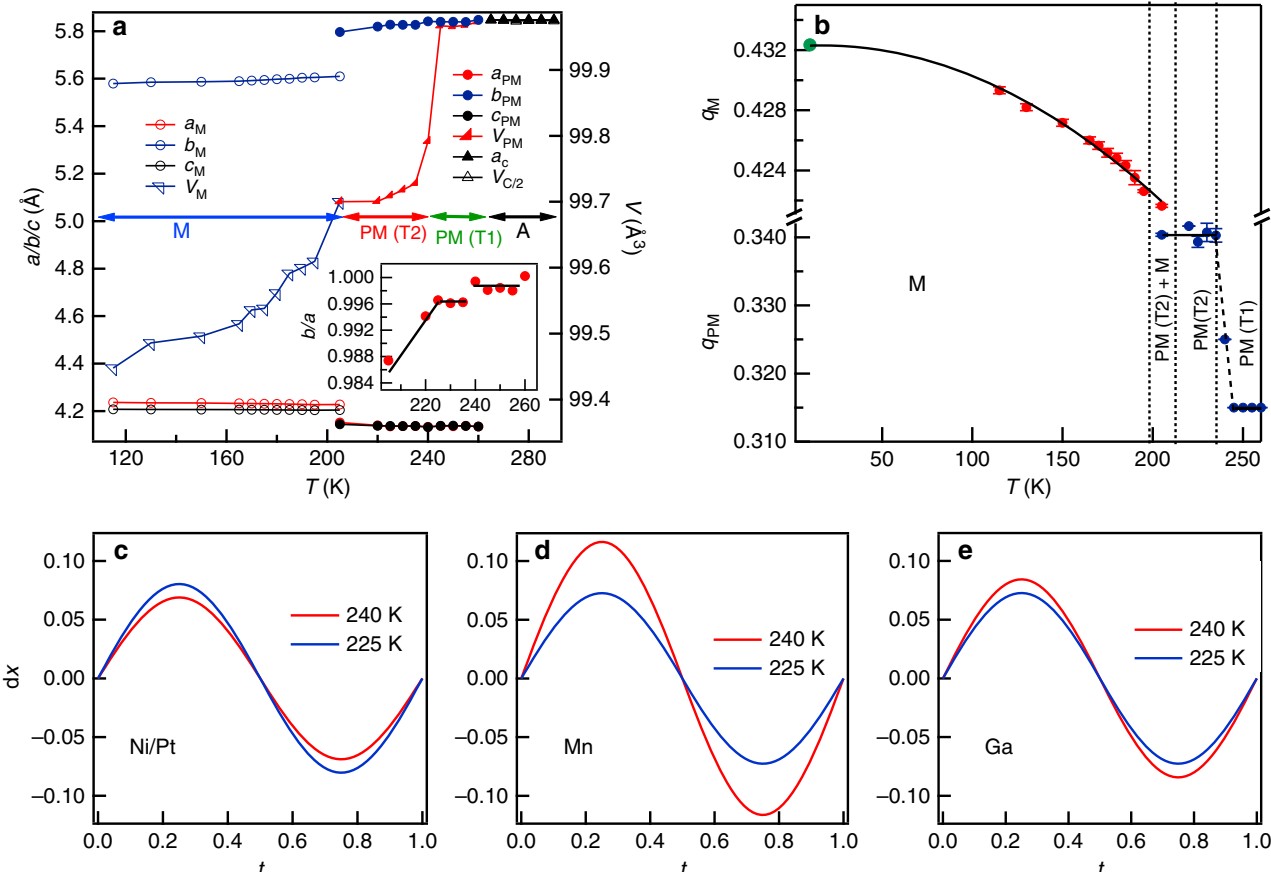

**Fig. 6** Temperature variation of the refined parameters. The parameters obtained from Rietveld refinements are plotted in **a** lattice parameters *a*, *b*, *c*, and volume in the austenite (A), 3M-like premartensite (PM), and the 7M-like martensite (M) phase regions for the cooling cycle. The volume of cubic phase is scaled with 1/2 for comparison. Inset shows the variation of *b/a* ratio (Bain distortion) with temperature in PM (T1) and PM (T2). Subscript c, PM, and M are used for austenite cubic, PM, and M phases, respectively. **b** Variation of magnitude of modulation vector (**q**) as a function of temperature obtained from superspace Rietveld refinements during cooling; **c**–**e** show the comparison of displacement (*dx*) with *t* parameter between PM (T1) at 240 K and PM (T2) at 225 K

this transition as well. The *b/a* ratio shown in the inset of Fig. 6a also exhibits discountinuous change at temperatures corresponding to the different phases. We believe that the formation of the PM (T1) phase in the cubic austenite matrix, followed by robust Bain-distorted PM (T2) phase in the PM (T1) matrix and finally the fully Bain-distorted martensite phase in the PM (T2) phase is kinetically more favorable for maintaining an invariant habit plane all through the transition due to the gradual evolution of the Bain distortion.

The displacement of atoms obtained from the Rietveld refinements (Supplementary Discussion and Supplementary Table 1) for PM(T1) at 240 K and PM (T2) at 225 K phases are compared in Fig. 6c–e, which clearly indicates a change in the atomic displacement between two phases. Since, both the PM phases belong to the same superspace group *Immm* (00γ) s00 with nearly 3M-like incommensurate modulation, this is an isostructural phase transition where the atomic positions shift without changing the Wyckoff site symmetries and the over all space group. The 3D approximant structures of PM (T1) and PM (T2) phases are compared in Supplementary Table 2. The discontinous change in the unit cell parameters at the isostructural phase transition temperature clearly reveals the presence of strong spin lattice (magnetoelastic) coupling. Isostructural phase transitions are rather rare and have been reported for example in the past during electronic transitions in chalcogenides under high pressure[53], in multiferroics across the

magnetic phase transition leading to excess spontaneous polarization due to magnetoelectric coupling[54–57], between two ferroelectric phases[58] and between two antiferrodistorted structures in CaTiO₃[59]. Our results provide evidence for an isostructural phase transition from a nearly cubic-like PM phase to a robust Bain-distorted PM phase in a magnetic SMA.

An incommensurate phase is often perceived to undergo a transition to the lock-in commensurate phase at low temperatures in the ground state[60]. However, our recent high resolution SXRPD study on the stochiometric Ni₂MnGa composition reveals that modulation wave vector (**q**) shows a smooth analytical behavior down to 5 K and there is no evidence for any devilish plateau or commensurate lock-in phase[27]. Since the PM phase in Ni₁.₉Pt₀.₁MnGa shows robust Bain distortion not present in the stoichiometric composition, it is intersting to verify the possibility of the lock-in phase in this system. For this, we carried out Rietveld refinement of the structures at various temperatures to determine the modulation wave vector **q** as a function of temperature using SXRPD data down to 110 K. In addition, we also carried Le Bail refinement using high resolution neutron powder diffraction data at 10 K to verify whether the incommensurate phase would lock-into a commensurate phase in the ground state. The temperature variation of the modulation wave vector **q** is shown in Fig. 6b during cooling. A jump *q* (magnitude of **q**) is clearly observed at the 3M modulated PM (T1) to PM (T2) isostructural phase transtion while both phases continue to

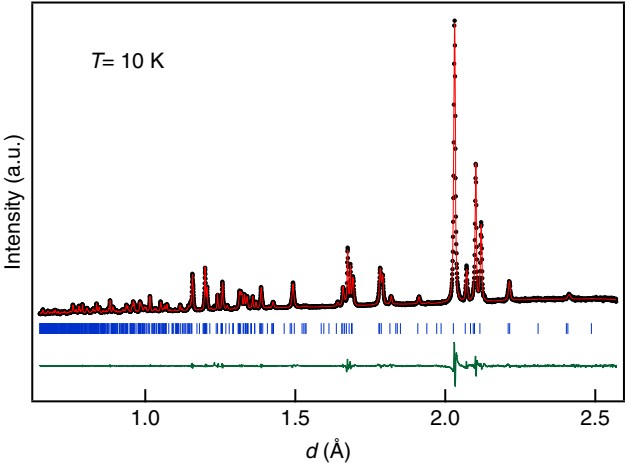

**Fig. 7** Le Bail fits for the neutron diffraction patterns at 10 K. The experimental data, fitted curve and the residue are shown by circles (black), red continuous line, and green continuous line, respectively. The tick marks (blue) represent the Bragg peak positions

remain incommensurate. Furthermore, the incommensurate 3M-like PM(T2) to the 7M-like incommensurate martensite phase transition is also accompanied with a discontinuous change in the modulation wave vector confirming the first-order nature of this transition as well. This is consistent with the observation of discontinuous change in volume and phase coexistence around $T_M$. It is interesting to note that the $q$ for PM (T1) is slightly less than the commensurate value of 0.33 whereas the $q$ of PM (T2) is slightly greater than 0.33. This implies that the modulation vector remains incommensurate in both the PM phases. However, the possibility of $q$ passing through 0.33 (1/3) commensurate value in the discontinous jump region with a plateau for $q = 1/3$ cannot be ruled out. From the refinement of the structure using the neutron diffraction data at 10 K (Fig. 7), we obtained the lattice parameters as $a = 4.24452(9)$, $b = 5.55035(11)$, $c = 4.20861(8)$ and an incommensurate value of the modulation vector $q = 0.43238 \pm 0.00021$. This value of **q** confirms that the martensite structure remains incommensurate down to 10 K and shows a smooth analytical behavior in the entire temperature range of stability of the martensite phase without any intermediate or ground state commensurate lock-in phase in Pt-substituted Ni₂MnGa similar to that in the undoped composition[27]. Thus, despite the robust Bain distortion, the ground state remains incommensurate.

**Theory**. To understand the role of Pt in causing robust Bain distortion in the PM phase, we have performed first principles calculations of the stoichiometric $Ni_2MnGa$ and $Ni_{1.83}Pt_{0.17}MnGa$ (shown in Fig. 8). For $Ni_{1.83}Pt_{0.17}MnGa$ (orange squares in Fig. 8a), our calculations reveal a pronounced energy minimum at $c/a \sim 0.993$, which indicates Bain distortion in the PM phase of this composition. This Bain distortion is caused by Pt, since a similar minimum is not present in $Ni_2MnGa$ (blue circles in Fig. 8b). Since Pt is non-magnetic, it is unlikely to cause direct magnetoelastic coupling, which is otherwise typical for NiMn-based Heusler alloys. Instead, both our measurements and calculations indicate that the substitution of larger Pt atoms in place of smaller Ni atoms increases the optimum volume of the alloy. In order to clearly understand the role of volume, we have calculated the total energy of $Ni_{1.83}Pt_{0.17}MnGa$ as function of $c/a$ ratio at the lower optimum volume of $Ni_2MnGa$ (blue circles in. Fig. 8a). It can be seen that lowering the volume yields a flat energy plateau with no clear minimum present at $c/a \sim 0.993$. This clearly shows

that the volume expansion due to Pt substitution causes the robust Bain distortion in $Ni_{1.83}Pt_{0.17}MnGa$.

To further check this, we calculated energies as function of $c/a$ ratios for $Ni_2MnGa$ at the two different volumes. Our calculations reveal again an energy plateau with reduced value for $Ni_2MnGa$ at its equilibrium volume without an energetic preference for $c/a \sim 0.993$ (blue circles in Fig. 8b), but does not exclude the possibility to observe the Bain-distorted phase in unsubstituted composition. However, at the larger equilibrium volume of $Ni_{1.83}Pt_{0.17}MnGa$, the Bain-distorted PM state has lower energy at $c/a \sim 0.993$, as can be clearly seen from Fig. 8b. Interestingly, we find that the reduced magnetization at finite temperature captured here through fixed-spin moment calculations[20] stabilizes the cubic PM phase (Fig. 8b). Hence, while the Pt substitution at the Ni site modifies the energy plateau found for $Ni_2MnGa$ through the enhanced volume and stabilizes the robust Bain-distorted PM phase at lower temperatures, the finite temperature decrease in magnetization will stabilize the cubic PM phase at higher temperatures. This explains the experimentally observed sequence of structural transitions in the PM phase.

In Fig. 8c, we present calculated magnetic moments for $Ni_2MnGa$ at the two different volumes. It can be clearly seen that the magnetic moments slightly (by ~ 1%) increase at the higher volume of $Ni_{1.83}Pt_{0.17}MnGa$ (orange squares) as compared to those obtained at the equilibrium volume of $Ni_2MnGa$ (blue circles). The more important observation is a discontinuous jump in magnetic moment for $c/a \sim 0.993$ at higher volume. This discontinuous jump is linked with the sudden change of the energy state at the same $c/a$ ratio (Fig. 8b).

## Discussion

In this work, we have presented evidence for robust Bain distortion in the PM phase of $Ni_2MnGa$ as a result of 10% Pt substitution using high resolution SXRPD. This finding is significant for the field of SMAs in general and MSMAs in particular. Our results show that the austenite to martensite phase transition in $Ni_{1.9}Pt_{0.1}MnGa$ involves three intermediate steps. The first transition is from the austenite to 3M-like incommensurate modulated PM phase PM (T1), which preserves the cubic symmetry of the austenite peaks similar to that in the stoichiometric $Ni_2MnGa$. The second transition is an isostructural transition from 3M-like incommensurate modulated PM phase (PM (T1)) with negligible Bain distortion to another 3M-like incommensurate modulated PM phase (PM (T2)) with robust Bain distortion without any change in the superspace group symmetry. Then this new PM (T2) phase finally transforms to the 7M-like incommensurate modulated martensite (M) phase. Our results thus suggest that Pt substitution leads to gradual increase in Bain distortion which can facilitate the invariant habit plane requirement all through the three transitions until the martensite phase is formed. Our first principles calculations show that volume expansion due to Pt substitution provides the driving force for the robust Bain distortion of the PM (T2) phase in Pt doped $Ni_2MnGa$ at lower temperatures while the reduced magnetization stabilizes the cubic-like PM (T1) phase at higher temperatures. Our findings provide an opportunity to critically reevaluate the applicability of the two main theories suggested for the origin of the modulation. The first one is related to the adaptive phase concept, which considers the ground state of the martensite phase to be $L1_0$ type non-modulated tetragonal structure resulting from the lattice deformation strain (Bain distortion) of the cubic austenite phase. In this model the modulated structure is a metastable phase formed due to kinetic reasons through nanotwinning of the Bain-distorted austenite structure. This model does not support the existence of the intermediate PM phase and treats the martensite

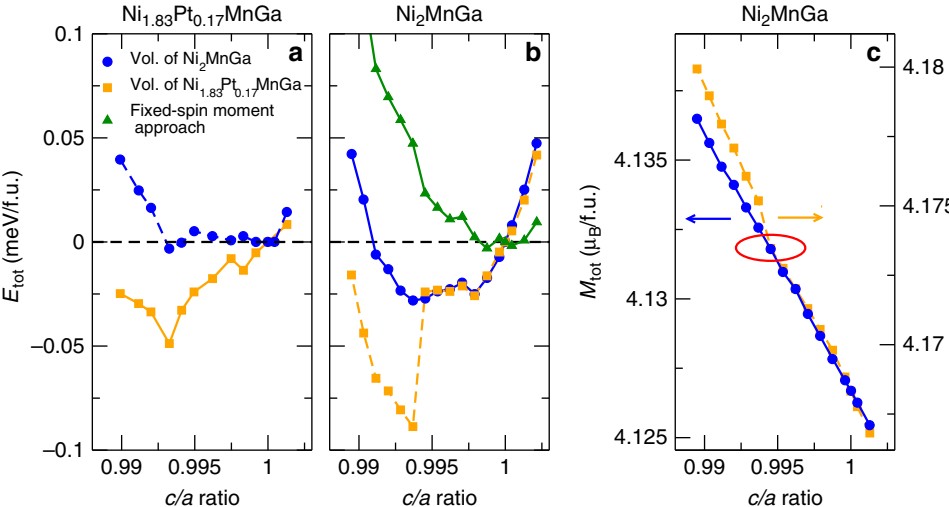

**Fig. 8** Calculations for energy vs $c/a$ variation. Variation of the total energy at $T = 0$ K with $c/a$ ratio for the 3M premartensite phase of **a** $Ni_{1.83}Pt_{0.17}MnGa$ and **b** $Ni_2MnGa$. Square symbols correspond to calculations performed at the equilibrium volume of $Ni_{1.83}Pt_{0.17}MnGa$, while circles denote calculations performed at the equilibrium volume of $Ni_2MnGa$. The energy computed with fixed-spin moment approach for $Ni_2MnGa$ at the equilibrium volume of $Ni_{1.83}Pt_{0.17}MnGa$ is shown by triangles. **c** The variation of magnetic moment with $c/a$ ratio for $Ni_2MnGa$ at its optimum volume (circles) and at the optimum volume of $Ni_{1.83}Pt_{0.17}MnGa$ (squares). The red ellipse marks a discontinuous jump of the magnetization (yellow curve)

phase as a kinetically stabilized state to overcome the lattice deformation shear causing the Bain distortion[32]. The other theory is based on a displacive modulation concept, where the origin of modulation is related to a TA2 soft acoustic phonon mode of the austenite phase[39] that gets coupled to the charge density wave in the PM phase[34] with opening of a pseudogap at the Fermi surface. In this model, an incommensurate modulated martensite phase can result from the incommensurate modulated PM phase[27], both of which are stable thermodynamic phases in their respective range of temperatures. Our results clearly show that the incommensurate martensite phase with larger Bain distortion appears in steps from the Bain-distorted PM phase, which supports the soft phonon mode based model as the likely origin of modulation in $Ni_2MnGa$. The understanding of the origin of modulation in these alloys provides a pathway to design new MSMAs.

Before we close, we would like to briefly mention the likely implications of our results on the functional properties of $Ni_2MnGa$. In view of the fact that large MFIS observed in $Ni_2MnGa$ is intimately linked with the existence of the modulated structure, the gradual evolution of modulation wave vector in going from PM (T1) to PM (T2) to martensite phase may influence the magnitude of MFIS. MSMAs also exhibit magnetocaloric effect (MCE) but the off-stoichiometric Ni–Mn–X (X = Ga, In, Sn, Sb) show larger MCE as compared to $Ni_2MnGa$[18, 61]. However, the off-stoichiometric alloys show larger hysteresis leading to irreversible MCE, which is a major drawback for any practical application[62, 63]. Since the gradual evolution of the Bain distortion can facilitate the emergence of an invariant habit plane, we believe that such alloys may exhibit better reversibilty due to lower hysteresis.

## Methods
**Sample preparation and characterization**. A polycrystalline sample of $Ni_{1.9}Pt_{0.1}MnGa$ was prepared by the standard arc melting technique. The actual composition of the sample was checked by energy dispersive analysis of X-rays to be $Ni_{1.9}Pt_{0.08}Mn_{1.04}Ga_{0.98}$. The characteristic (magnetic and structural) transition temperatures were obtained by the ac-susceptibility measurements as a function of temperature using a SQUID-VSM magnetometer.

**Synchrotron X-ray and neutron powder diffraction and analysis**. High resolution synchrotron powder X-ray diffraction (SXRPD) measurements were performed at the P02 beamline in Petra III, Hamburg, Germany using a wavelength of 0.20712 Å. The time of flight neutron diffraction data were obtained at the ISIS facility (UK) using the high resolution powder diffractometer HRPD at 10 K. For SXRPD and neutron diffraction measurements, the powder samples obtained by crushing the as-cast ingots were annealed at 773 K under a high vacuum of $10^{-5}$ mbar for 10 h to remove the residual stresses introduced during grinding[62, 64]. The analysis (Le Bail and Rietveld) of diffraction data were performed using $(3 + 1)$ D superspace group approach[65–67] with the JANA2006 software package[68].

**Theoretical calculation**. For theoretical calculations, we have chosen a slightly different Pt concentration as compared to the experimental composition in order to minimize computational time. Nevertheless, we demonstrate that the physical mechanism responsible for the Bain distortion of the PM phase does not depend on the precise Pt concentration. Further, we have assumed a commensurate 3 M structure for the PM phase and have also neglected finite temperature entropic contributions. The spin polarized density functional theory (DFT) calculations have been performed for a 24-atom supercell using Vienna Ab-Initio Simulation Package[69]. Relaxed geometries and atomic positions are determined using the projector-augmented wave method[70]. The exchange-correlation functional is described by the generalized gradient approximation of Perdew–Burke–Ernzerhof[71]. To avoid errors in calculated total energy because of artificial Pulay stress arising due to incomplete basis set[72], an energy cutoff of 700 eV is used for the plane wave basis and the independence of this choice (achieved convergence in the total energy ~0.01 meV/f.u.) has been checked. For the Brillouin zone sampling, a k-point grid of $8 \times 24 \times 18$ is used. Convergence criteria for the electronic structure and the largest residual forces of respectively $10^{-8}$ eV and $10^{-4}$ eV/Å are used. Such strict choice of cutoff parameters and convergence criteria resulted in DFT energies with error <0.003 meV/atom. Finite temperature magnetic effects have been modeled by constraining the total moment per cell by means of the fixed-spin moment approach[73].

**Data availability**. The data that support the findings of this study are available upon request from S.S.

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

## Acknowledgements

The work was financially supported by the ERC AG 291472 'IDEA Heusler! and Heu-Mem project within the European network M-era.net. S.S. thanks S.R. Barman and A. Chakraborty for useful discussion and J. Bednarcik for help in SXRPD measurements. The support from DST under its India-DESY scheme through Jawaharlal Nehru Centre for Advanced Scientific Research is acknowledged. Experiments at the ISIS Pulsed Neutron and Muon Source were supported by a beamtime allocation from the Science and Technology Facilities Council. B.D. and T.H. gratefully acknowledge Deutsche Forschungsgemeinschaft (DFG) for their funding within the priority program SPP1599. S.S. thanks Alexander von Humboldt foundation, Germany. D.P. acknowledges support from Science and Engineering Research Board of India for financial support through the award of J.C. Bose Fellowship.

## Author contributions

S.S. proposed the problem. C.F. and D.P. supervised the project. S.S. prepared the sample. P.D. performed compositional analysis. S.S. performed magnetic and synchrotron experiments. S.S., M.G.Z., and A.S.G. did neutron diffraction experiment. S.S. and D.P. analyzed the experimental data. B.D. performed theoretical calculations. S.W.D.S., S.C., and T.H. provided inputs for theory. S.S. and D.P. wrote the manuscript with substantial feedback from all co-authors.

## Additional information

**Competing interests:** The authors declare no competing financial interests.

