## [Peer Review File · Nature Communications]

Reviewers' Comments:

Reviewer #1 (Remarks to the Author)

The article by Singh et al. reports on the discovery of two premartensite phases in a Ni₂MnGa alloy with 10% Pt substitution. The interesting new observation is that there occurs a second premartensite phase with intermediate Bain distortion, specifically for the studied alloy. However, the authors have failed to explain the detailed mechanism of the transformation and also failed to put this information in broader perspective.

First, what is the atomic mechanism for the two transformations? What are the different atomic displacements? Are the atomic displacements same or different? This kind of information can truly provide new insights into the physics of the problem. The authors did carry out Rietveld refinement, but these details are not available from their analysis.

Secondly, it is not clear what new information is brought to table regarding the martensitic phase transformation phenomenon. The authors state that their observations indicate support for the soft phonon mode based model. But this has already been shown in previous studies, such as ref. 40, also Pramanick Phys. Rev. B 85, 144412 (2012).

Thirdly, it is not clear what is the broader perspective. The authors have failed to convincingly describe the role of Pt in stabilizing the premartensite phase. What is the insight, with regard to alloy design, that is being brought. For example, what other alloying elements can have a similar effect.

Finally, the authors have stated in the abstract that the results gives better understanding of functional properties of the alloy. This is not clear. What properties are being described and what role the premartensite phase plays in them?

Therefore, while the result for an additional premartensite phase with stable Bain distortion in itself is interesting, the underlying physics behind it has not been described adequately and the broader implication for this is not clear. In view of this, I regret I cannot recommend publication of the article in Nature Communications in its present form.

Reviewer #2 (Remarks to the Author)

The manuscript presents a detailed study of the sequence of transformations in a magnetic shape memory alloy of Ni₂MnGa with 10% of Pt. The authors evidence, by means of high resolution synchrotron X-ray diffraction, the presence of a premartensitic phase with a Bain distortion. Even though the characteristics of the intermediate phase between austenite and martensite has been extensively studied in Ni-Mn-Ga alloys, there is still controversy about the origin of the modulation shown by the low temperature phases.

In this sense, the present work shows a clear evidence of a new transformation sequence: austenite--undistorted premartensite---distorted premartensite---martensite, not reported until now. This finding can provide important information about the transformation mechanism and the origin of the martensitic phase modulation. Taking this into account, the paper deserves publication in Nature Communications. In order to be published the following questions should be addressed.

In a previous work, ref 28, on the Ni₂MnGa system, only the undistorted premartensite phase is observed in the transformation sequence. If the presence of a Bain distorted premartensitic phase is more favorable to maintain an invariant habit plan, could one expect the presence of this phase in Ni₂MnGa even in a very narrow temperature range?

Could the observed transformation sequence be generalized to all systems with premartensite

phase? Or is it the result of the modification in the magnetoelastic coupling due to the disorder and volume change induced by the Pt addition?

The authors describe some differences in the magnetic measurements between NiPtMnGa and Ni₂MnGa, i.e. the absence of dip in the susceptibility at the transition temperature of the premartensitic phase and a huge drop in the susceptibility linked to this phase. Could the authors give some explanation to these differences?

Taking into account the large susceptibility and volume changes, fig. 1 and fig 6, Why the authors consider type II premartensite and not an intermediate martensitic phase with modulation 3M?

There is no figure 2-e. Please correct figure 2, figure caption and corresponding text.

Please, check some typing errors, sort references "which is incommensurate with 3M like modulation.26,28,27,29,31,51,52"

Reply to reviewers' comments:

Reviewer #1 (Remarks to the Author):

Comment: The article by Singh et al. reports on the discovery of two premartensite phases in a Ni₂MnGa alloy with 10% Pt substitution. The interesting new observation is that there occurs a second premartensite phase with intermediate Bain distortion, specifically for the studied alloy. However, the authors have failed to explain the detailed mechanism of the transformation and also failed to put this information in broader perspective.

Reply: We appreciate referee's comments about our manuscript "The interesting new observation is that there occurs a second premartensite phase with intermediate Bain distortion". In the following, we have tried our best to provide point-by-point response to the referee's comments/suggestions.

Comment 1: First, what is the atomic mechanism for the two transformations? What are the different atomic displacements? Are the atomic displacements same or different? This kind of information can truly provide new insights into the physics of the problem. The authors did carry out Rietveld refinement, but these details are not available from their analysis.

Reply: We have now provided the detailed results of Rietveld refinement (atomic displacement, atomic position, symmetry, lattice parameters etc.) for the two premartensite phases (i) PM (T1), where the cubic symmetry is maintained with no evidence of Bain distortion and (ii) PM (T2), which exhibits robust Bain distortion. The results clearly indicate that PM (T1) to PM (T2) transformation is isostructural type as the crystal symmetry and Wyckoff positions are same for both the phases and only the atomic displacements are different. We have compared the atomic displacement in the two premartensite phases in the revised manuscript in Fig.6 c-e. Details of Rietveld refinement results are provided in the supplementary file of the revised manuscript.

Comment 2: Secondly, it is not clear what new information is brought to table regarding the martensitic phase transformation phenomenon. The authors state that their observations indicate support for the soft phonon mode based model. But this has already been shown in previous studies, such as ref. 40, also Pramanick Phys. Rev. B 85, 144412 (2012).

Reply: The premartensite phase, which has been extensively studied in Ni-Mn-Ga magnetic shape memory alloys appears in between the parent austenite and martensite phases and has always been considered as a precursor effect with preserved symmetry of the high temperature cubic phase. This is because there is no evidence of symmetry breaking or in other words any evidence for Bain distortion in the premartensite phase. In the present work we have presented direct evidence for robust Bain distortion (i.e., broken symmetry) in the premartensite phase. This shows that premartensite is an independent thermodynamic phase, which further transforms to the martensite phase. Although the modulated premartensite phase has been observed in many alloys, a clear relationship between the modulated premartensite phase and the origin of the modulation

of martensite phase is still an open issue. The present work shows clear evidence that Bain distorted martensite phase results from the Bain distorted premartensite phase, which has been never reported earlier. This observation also indicates that adaptive modulations of martensites are not applicable for the present alloy as no direct interface is formed between cubic austenite and tetragonal martensite. Therefore present finding provide clear information about the transformation mechanism and the origin of the modulation in the martensitic phase as soft phonon mode based model. The results of ref. 40 and also Pramanick Phys. Rev. B 85, 144412 (2012) have not addressed this issue of the premartensite phase, which gives direct evidence about the phase transformation sequence and hence the origin of modulation.

Reviewer2 has also mentioned this in his/her comment “*Even though the characteristics of the intermediate phase between austenite and martensite has been extensively studied in Ni-Mn-Ga alloys, there is still controversy about the origin of the modulation shown by the low temperature phases. In this sense, the present work shows a clear evidence of a new transformation sequence: austenite--undistorted premartensite---distorted premartensite---martensite, not reported until now. This finding can provide important information about the transformation mechanism and the origin of the martensitic phase modulation.*”

Comment 3: Thirdly, it is not clear what is the broader perspective. The authors have failed to convincingly describe the role of Pt in stabilizing the premartensite phase. What is the insight, with regard to alloy design, that is being brought. For example, what other alloying elements can have a similar effect.

Reply: To understand the exact mechanism responsible for the stabilization of Bain distorted premartensite in $\text{Ni}_{1.9}\text{Pt}_{0.1}\text{MnGa}$, we have performed new *ab initio* calculations (please see modified Fig.8 and the corresponding text on pages 14-16 in the revised version of the manuscript). Our calculations indicate that volume expansion due to larger Pt atoms is responsible for the Bain distortion in $\text{Ni}_{1.83}\text{Pt}_{0.17}\text{MnGa}$ (slightly different composition from experiment is chosen to minimize computational effort). We have demonstrated this by calculating the energy as a function of c/a for $\text{Ni}_{1.83}\text{Pt}_{0.17}\text{MnGa}$ at the lower optimum volume of Ni_2MnGa . These calculations at the lower volume do not show any Bain distortion for the premartensite phase. To establish the mechanism more clearly we have calculated the premartensite phase of the parent compound Ni_2MnGa at the larger optimum volume of $\text{Ni}_{1.83}\text{Pt}_{0.17}\text{MnGa}$. Our calculations clearly show that a Bain distorted premartensite phase is also stable for Ni_2MnGa at the larger volume while no indications of Bain distortion can be seen for the parent compound at its equilibrium volume. Further, our investigations on the magnetic properties show that the magnetization in Ni_2MnGa increase at the larger volume. This indicates that higher magnetization at the larger volume helps in symmetry breaking and producing robust Bain distortion in the premartensite phase due to strong magnetoelastic coupling. Since the volume of $\text{Ni}_{1.83}\text{Pt}_{0.17}\text{MnGa}$ ($\text{Ni}_{1.9}\text{Pt}_{0.1}\text{MnGa}$ in experiments) is larger than that of Ni_2MnGa , similar mechanism of magnetoelastic coupling is also responsible for the Bain distortion of its premartensite phase.

The stability of the premartensite phase is extremely sensitive to alloy composition and external fields such as pressure, temperature and magnetic fields. In the present case, Pt is non-magnetic

and has same number of valence electrons as that of Ni. Hence, Pt influenced the transformation process only due to its larger atomic size. Substitution of a different element will most likely change the valence electron concentration, atomic size and possibly magnetic properties. Hence, it is extremely difficult to make a general statement about the effect of any alloying element. While the present study shows the effect of volume, systematic studies in future is required to understand the effect of other alloying elements.

Comment 4: Finally, the authors have stated in the abstract that the results give better understanding of functional properties of the alloy. This is not clear. What properties are being described and what role the premartensite phase plays in them?

Reply: The stoichiometric Ni_2MnGa is an important system for application as magnetic actuator due to its large magnetic field induced strain (MFIS). This system exhibits not only a large MFIS but also magnetocaloric effect (MCE) and topological spin textures. Besides Ni_2MnGa , the off-stoichiometric Ni-Mn-Ga together with other Ni-Mn-X (X= In, Sn, Sb) based magnetic shape memory alloys (MSMA's) are extensively investigated due to their large MCE. However, in contrast to Ni_2MnGa , they show a large hysteresis and irreversible behavior, which is a major drawback for their use in practical application[Phys. Rev B 92, 020105, (2015)]. It is interesting to note that in Ni_2MnGa austenite phase transforms first to the premartensite phase, which further transforms to the modulated martensite phase. On the other hand, the off stoichiometric Ni-Mn-X (X= In, Sn, Sb) directly transform from austenite to the martensite phase without showing any precursor effect [Nature Mater. 4, 450 (2005)]. In the Pt substituted Ni_2MnGa (studied here), the martensite phase originates from the larger Bain distorted premartensite phase suggesting that the Bain distortion appears in steps to facilitate the emergence of an invariant habit plane. Therefore, the premartensite phase may be able to adapt low inputs of magnetic or elastic energy change without undergoing irreversible effects. However, the main emphasis of this paper is to show that robust Bain distortion is possible in premartensite phase also and the consequence of this in the functional properties needs to be investigated.

In view of the fact that large MFIS is observed in Ni_2MnGa is intimately linked with the existence of the modulated structure, which exhibits austenite-premartensite(PM)-martensite phase transformation sequence the successive change in modulation wave vector is going from PM (T1)--PM (T2)--martensite is likely to affect the observed MFIS. In the conclusion of manuscript we have mentioned "This basic understanding of the origin of modulation in these alloys provides a pathway to design new MSMA's since the large MFIS in Ni_2MnGa is intimately linked with the existence of the modulated structure." We have also modified now the related sentence in the abstract as "Our work is a major breakthrough in the field of MSMA's as it gives better understanding of the premartensite phase and its relation to the martensite phase transition, which is necessary for engineering the functional properties of these alloys." However, if the reviewer considers it necessary, we will include an elaborative discussion in the manuscript.

Comment 5: Therefore, while the result for an additional premartensite phase with stable Bain distortion in itself is interesting, the underlying physic behind it has not been described

adequately and the broader implication for this is not clear. In view of this, I regret I cannot recommend publication of the article in Nature Communications in its present form.

Reply: We again thank referee for positive remark about our findings. We have now taken care of all the concerns of the referee about our manuscript and included new results of Rietveld refinements and first principles calculation, which provides the explanation for the stabilization of Bain distorted premartensite phase, observed for Pt substitution. We believe that referee will find our revised manuscript suitable for publication in Nature Communications.

Reviewer #2 (Remarks to the Author):

Comment: The manuscript presents a detailed study of the sequence of transformations in a magnetic shape memory alloy of Ni₂MnGa with 10% of Pt. The authors evidence, by means of high resolution synchrotron X-ray diffraction, the presence of a premartensitic phase with a Bain distortion. Even though the characteristics of the intermediate phase between austenite and martensite has been extensively studied in Ni-Mn-Ga alloys, there is still controversy about the origin of the modulation shown by the low temperature phases. In this sense, the present work shows a clear evidence of a new transformation sequence: austenite--undistorted premartensite---distorted premartensite---martensite, not reported until now. This finding can provide important information about the transformation mechanism and the origin of the martensitic phase modulation. Taking this into account, the paper deserves publication in Nature Communications. In order to be published the following questions should be addressed.

Reply: We thank referee for his extremely positive comments about our manuscript and his recommendation for publication of our manuscript in Nature Communications. We are providing answers to referee's questions below.

Comment 1: In a previous work, ref 28, on the Ni₂MnGa system, only the undistorted premartensite phase is observed in the transformation sequence. If the presence of a Bain distorted premartensitic phase is more favorable to maintain an invariant habit plane, could one expect the presence of this phase in Ni₂MnGa even in a very narrow temperature range? Could the observed transformation sequence be generalized to all systems with premartensite phase? Or is it the result of the modification in the magnetoelastic coupling due to the disorder and volume change induced by the Pt addition?

Reply: This is a very interesting question raised by the referee which encouraged us to undertake first principles calculations for pure and Pt substituted Ni₂MnGa. These calculations reveal that premartensite phases with $c/a \sim 1$ and ~ 0.993 are nearly degenerate (see Fig. 8b) suggesting the possibility of observing Bain distorted phase in unsubstituted compositions also at least in a narrow temperature regime. We have mentioned this on page 15 that Pt substitution lifts this degeneracy through enhanced volume magnetization (see Fig 8a and c). However, to confirm the actual crystal structure and nature of transformation, one needs more sophisticated measurements with ultra high resolution systems. One cannot ignore this possibility in other systems showing

premartensite phase transition due to the observation of magnetoelastic effects at the premartensite phase transition. However, a detailed investigation of these systems using still higher resolution diffraction data will be useful to generalize the present finding. We hope our publication will encourage researchers to work in this direction.

Comment 2: The authors describe some differences in the magnetic measurements between NiPtMnGa and Ni₂MnGa, i.e. the absence of dip in the susceptibility at the transition temperature of the premartensitic phase and a huge drop in the susceptibility linked to this phase. Could the authors give some explanation to these differences?

Reply: Here we have tried to explain the change in ac- χ across the premartensite phase transition. In Ni₂MnGa a very small (nearly 0.5 %) drop in ac- χ at the premartensite phase (~260 K) during cooling from the austenite phase is observed [Ref.28, Fig.1]. This small drop in ac- χ is mentioned as dip. On the other hand, nearly 40% of the total drop in ac- χ has already occurred at 235K in the premartensite phase for the Ni_{1.9}Pt_{0.1}MnGa. This indicates larger magnetocrystalline anisotropy of the premartensite phase in Ni_{1.9}Pt_{0.1}MnGa. For better clarity we have now modified related sentences in the revised manuscript.

Comment 3: Taking into account the large susceptibility and volume changes, fig. 1 and fig 6, Why the authors consider type II premartensite and not an intermediate martensitic phase with modulation 3M?

Reply: We completely agree with the referee that Bain distorted premartensite phase should be considered as separate phase. However since this 3M Bain distorted phase (type II premartensite) arises from the 3M premartensite phase where the cubic symmetry is maintained and this phase further transforms to martensite phase, therefore in the present nomenclature scenario in the field we have termed it as Bain distorted premartensite phase.

Comment 4: There is no figure 2-e. Please correct figure 2, figure caption and corresponding text.

Reply: This has been corrected in the resubmitted version. We thank the referee for pointing it out.

Comment 5: Please, check some typing errors, sort references "which is incommensurate with 3M like modulation.26,28,27,29,31,51,52"

Reply: All these have been corrected in the resubmitted version. We thank the referee for pointing it out.

Reviewers' Comments:

Reviewer #1:

Remarks to the Author:

The authors have adequately addressed the concerns raised in the previous reviews. Their Rietveld refinement results and first-principles calculations now convincingly establish the mechanism for transition from austenite-to-martensite phase, through pre-martensite phases. They have also adequately addressed the broader impact of the study, which is the central role played by volume expansion towards Bain distorted premartensite phase. I will recommend the revised manuscript for publication in Nature Communications.

The authors could consider adding a short paragraph, summarizing their response to comment 4 from Reviewer#1

Reviewer #2:

Remarks to the Author:

I think the authors have adequately addressed the concerns raised during the review and have added some results to the original set to complete and support the discussion and conclusions. Overall, the current version of the manuscript is suitable to be published in Nature Communications.

REVIEWERS' COMMENTS:

Reviewer #1 (Remarks to the Author):

The authors have adequately addressed the concerns raised in the previous reviews. Their Rietveld refinement results and first-principles calculations now convincingly establish the mechanism for transition from austenite-to-martensite phase, through pre-martensite phases. They have also adequately addressed the broader impact of the study, which is the central role played by volume expansion towards Bain distorted premartensite phase. I will recommend the revised manuscript for publication in Nature Communications.

The authors could consider adding a short paragraph, summarizing their response to comment 4 from Reviewer#1

Reply: We thanks referee for recommendation of our manuscript for publication in Nature Communication. As suggested by the referee we have now included a separate paragraph in the discussion section.

Reviewer #2 (Remarks to the Author):

I think the authors have adequately addressed the concerns raised during the review and have added some results to the original set to complete and support the discussion and conclusions. Overall, the current version of the manuscript is suitable to be published in Nature Communications.

Reply: We thanks referee for recommendation of our manuscript for publication in Nature Communications.